

# Diversity and distribution of orchid bees (Hymenoptera: Apidae, Euglossini) in Belize

Kevin M. O'Neill[1], Ruth P. O'Neill[2], Casey M. Delphia[2], Laura A. Burkle[3] and Justin B. Runyon[4]

[1] Department of Land Resources and Environmental Sciences, Montana State University, Bozeman, MT, United States

[2] Department of Plant Sciences and Plant Pathology, Montana State University, Bozeman, MT, United States

[3] Ecology Department, Montana State University, Bozeman, MT, United States

[4] Rocky Mountain Research Station, U.S.D.A., United States Forest Service, Bozeman, MT, United States

Corresponding author
Kevin M. O'Neill,
koneill@montana.edu

## ABSTRACT

**Background:** Orchid bees are abundant and widespread in the Neotropics, where males are important pollinators of orchids they visit to collect fragrant chemicals later used to court females. Assemblages of orchid bees have been intensively surveyed in parts of Central America, but less so in Belize, where we studied them during the late-wet and early-dry seasons of 2015–2020.

**Methods:** Using bottle-traps baited with chemicals known to attract a variety of orchid bee species, we conducted surveys at sites varying in latitude, historical annual precipitation, elevation, and the presence of nearby agricultural activities. Each sample during each survey period consisted of the same number of traps and the same set of chemical baits, their positions randomized along transects.

**Results:** In 86 samples, we collected 24 species in four genera: *Euglossa* (16 species), *Eulaema* (3), *Eufriesea* (3), and *Exaerete* (2). During our most extensive sampling (December 2016–February 2017), species diversity was not correlated with latitude, precipitation, or elevation; species richness was correlated only with precipitation (positively). However, a canonical correspondence analysis indicated that species composition of assemblages varied across all three environmental gradients, with species like *Eufriesea concava*, *Euglossa imperialis*, and *Euglossa viridissima* most common in the drier north, and *Euglossa ignita*, *Euglossa purpurea*, and *Eulaema meriana* more so in the wetter southeast. Other species, such as *Euglossa tridentata* and *Eulaema cingulata*, were common throughout the area sampled. Mean species diversity was higher at sites with agricultural activities than at sites separated from agricultural areas. A Chao1 analysis suggests that other species should yet be found at our sites, a conclusion supported by records from adjacent countries, as well as the fact that we often added new species with repeated surveys of the same sites up through early 2020, and with the use of alternative baits. Additional species may be especially likely if sampling occurs outside of the months/seasons that we have sampled so far.

## INTRODUCTION

Orchid bees (Hymenoptera: Apidae, Euglossini) are endemic to the Neotropics where they comprise as much as 20–30% of bee species in some areas (*Roubik & Hanson, 2004*). Scientific interest in euglossines is partly motivated by the unusual form of pollination services provided by males. Males visit certain orchid species (Orchidaceae) solely to collect chemicals they use as pheromones during courtship (*Eltz, Roubik & Whitten, 2003*; *Eltz et al., 2007*; *Roubik & Hanson, 2004*; *Meisel, Kaufmann & Pupulin, 2014*; *Pokorny et al., 2017*). Males also collect chemicals at other sites, including plant sap, fungi, mammalian feces, and dead male orchid bees (*Roubik & Hanson, 2004*). Because many orchids produce volatile compounds collected by male orchid bees, but don't produce nectar or abundant pollen that could attract other pollinators, euglossines are critical components of Neotropical pollinator communities (*Ackerman, 1983a*; *Roubik & Hanson, 2004*). *Roubik & Hanson (2004)* estimate that about 10% of Central American orchid species are pollinated by male euglossines. In addition, other genera of orchids in Belize, as well as plants in more than 60 other families, are visited by both male and female orchid bees for nectar and pollen (*Roubik & Hanson, 2004*; *Meisel, Kaufmann & Pupulin, 2014*).

Orchid bee communities have been intensively studied in such places as the Atlantic Forest region of Brazil (*Nemésio, 2009*), Costa Rica, and Panama (*Roubik & Hanson, 2004*; *Vega-Hidalgo et al., 2020*). Farther north, quantitative faunal studies are less complete; *McCravy et al. (2016)* report on the relative abundance of euglossines in Cosuco National Park in northwest Honduras, and published species lists are available for Mexico (*Ayala, Griswold & Yanega, 1996*) and Nicaragua (*Hinojosa-Díaz & Engel, 2012*). An analysis by *Moure & Melo (2022)* indicated that the overall bee fauna, and that of orchid bees in particular, was not adequately documented across the Central America. For Belize, the number of all bee species listed in the comprehensive catalogue of *Moure & Melo (2022)* is only ~10% of the number of species they projected to be present based on its land area. *Schüepp, Rittiner & Entling (2012)* sampled orchid bees using baited Malaise traps in northeastern Belize where they recorded four species, but we have found no published studies that surveyed orchid bees quantitatively throughout the country or considered variables that might correlate with their distribution and diversity. Such information would be useful for understanding orchid-pollinator interactions, given that more than three dozen species of orchids in Belize (*Balick, Nee & Atha, 2000*; *Ames & Correll, 2012*) are in genera visited by male orchid bees for fragrance collection (*Roubik & Hanson, 2004*; *Meisel, Kaufmann & Pupulin, 2014*).

To better understand the diversity and distribution of orchid bees in Belize, we used baited traps with the overall objective of assessing its orchid bee fauna during the late-wet and early-dry seasons from late 2015 to early 2020. Here, we report on the frequency of orchid bee species captured in 86 multi-trap samples that included >7,800 trap-hours, and (at different times) 15 different chemical baits.

Our specific objectives were to (1) document the relative abundances of orchid bee species at different locations in the country and (2) using our largest set (December 2016–February 2017), provide an assessment of how the distribution and diversity of orchid bees correlate with latitude, elevation, mean annual precipitation, and the presence of agricultural activities within or adjacent to sample sites. The results provide a biogeographic data-base for future studies that could identify alterations in distribution and relative abundance of orchid bees that result from changes in climate (*Silva et al., 2015*; *Faliero, Nemésio & Loyola, 2018*; *Roubik et al., 2021*) and land-use patterns (*e.g.*, *Rincón et al., 1999*; *Schüepp, Rittiner & Entling, 2012*; *Briggs, Perfecto & Brosi, 2013*; *Nemésio, Santos & Vasconcelos, 2015*; *Oliveira, Pinto & Schlindwein, 2015*; *Botsch et al., 2017*; *Storck-Tonon & Peres, 2017*). For example, two recent studies examined trends in orchid bee communities over 40-year periods in Panama (*Roubik et al., 2021*) and Costa Rica (*Bravo et al., 2022*, who compared their results to those of *Janzen et al., 1982*). Quantitative distributional data could also be used to identify prime areas for orchid bee conservation (*Miranda et al., 2019*).

# MATERIALS AND METHODS

## Study sites

We sampled orchid bees in five of the six Belize Districts (Fig. 1), during what we will refer to as four survey "Periods": Period I) 15–30 December 2015 (10 sample sites at seven locations; Tables 1 and S1), Period II) 15 December 2016–27 February 2017 (44 sample sites at 18 locations; Tables 2 and S2), Period III) 4 October 2018–27 March 2019 (17 sample sites at 12 locations; Tables 3 and S3), and Period IV) 13 December 2019–2 January 2020 (15 sample sites at eight locations; Tables 4 and S4). We use the term "locations" to refer to general areas studied (*e.g.*, Black Rock or Punta Gorda), within which multiple "sites" were often sampled. Each site sampled during a survey period is given a location code (2–3 capital letters), followed by a numerical site designation (Tables 1–4), unless only one site was sampled at a location. The numerical designations are specific to each survey period, so a numeral given to a site in one survey period does not necessarily indicate the same physical location as a site with the same numeral in a different survey period. Latilong data for each site are found in in Tables S1–S4.

For each site, we determined its latitude, longitude, and elevation using *Google Earth Pro v. 7.12.2041 (2018)* (Tables S1–S4). Mean annual precipitation values represent median of yearly ranges for the period of 1951–2013 (*Belize.com Ltd., 2015*). Each site was given an ecosystem classification based on *Meerman & Sabido (2001)* (Tables S1–S4). The majority of the sites are classified as lowland broadleaf forests, but we also sampled submontane broadleaf forests at LC, submontane pine forests dominated by Caribbean pine (*Pinus caribaea* Morelet) at MPR, and lowland pine savannah sites of the Toledo and Stann Creek Districts where *P. caribaea* was also common. Within the ecosystem classes of *Meerman & Sabido (2001)*, the component plant species vary across Belize and there are ecosystem sub-classifications related to soil type. However, we used a simplification of their classification, and did not differentiate variants of particular general classes based on the dominant tree species present. For example, *Meerman & Sabido (2001)* have multiple

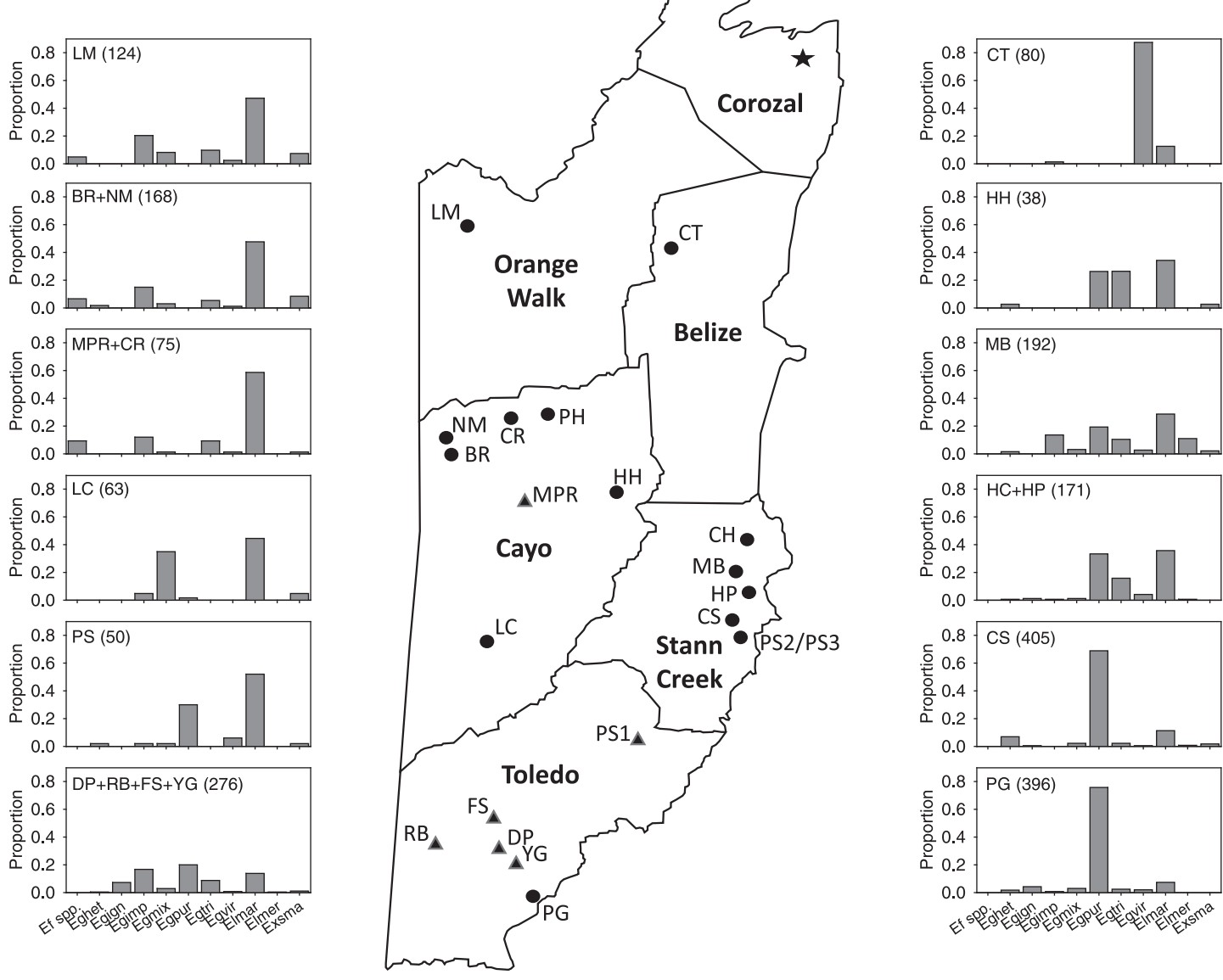

**Figure 1 Map of Belize show survey locations, with relative abundance of most abundant species in Period II.** Schematic map of mainland Belize with districts in bold print and showing all locations (circles) sampled from 2015–2020. At some locations, multiple samples were taken at different sites (Tables 1–4). Location labels correspond to those in Tables 1–4 and S1–S4. Histograms show proportions of some of the most abundant species in the 44 samples in Period II (species abbreviations are given in Table 5). Histograms include 37 of the 44 samples and 85% of specimens collected during that survey period.

sub-classes of lowland broadleaf evergreen seasonal forests, but we consolidate them into a single class.

We created a simple "Agricultural Presence Index" based on sites lacking (index = 0) or not having (index = 1) agricultural activities (*i.e.*, crop fields, citrus groves, or pastures) within or bordering the sampled areas. Some sites were purely agricultural (*i.e.*, the citrus groves of HH sites and old rice fields at DP). Agricultural areas were often closely associated with lowland broadleaf forest sites. The NM sites have a mosaic of forests, citrus

**Table 1 Sampling dates, species richness, and species diversity for Period I surveys.** Locations, sample sites, sampling dates, sample sizes, species richness, and species diversity for the Period I (2015) surveys. Latilong data, trap separation distances, and habitat characteristics for each site are given in Table S1.

| Location | Location/site code | Date sampled | Number of bees trapped | Species richness | Species diversity |
|---|---|---|---|---|---|
| Black Rock | BR | 15 Dec 2015 | 86 | 10 | 3.82 |
| Mountain Pine Ridge | MPR | 17 Dec 2015 | 81 | 9 | 4.19 |
| Chiquibul Road | CR | 18 Dec 2015 | 73 | 9 | 5.35 |
| Punta Gorda | PG1 | 21 Dec 2015 | 101 | 9 | 4.70 |
| | PG2 | 23 Dec 2015 | 78 | 11 | 4.96 |
| | PG3 | 24 Dec 2015 | 85 | 10 | 3.91 |
| Dump | DP | 25 Dec 2015 | 21 | 4 | 2.05 |
| Cockscomb Wildlife Sanctuary | CS1 | 27 Dec 2015 | 103 | 9 | 2.11 |
| | CS2 | 29 Dec 2015 | 109 | 11 | 2.08 |
| Hopkins | HP | 30 Dec 2015 | 20 | 6 | 4.35 |
| Mean | | | 75.7 | 8.8 | 3.75 |
| SE | | | 9.9 | 0.7 | 0.39 |

groves, and pastures. The sites at DP, HC, CR, FS, HP, and MB each included patches of forests or citrus groves, although traps at these locations were placed along roads or paths in forests or at edges of forests or patches of trees. Compared to sites without agricultural presence, those with agricultural presence (Table S2) did not differ in mean latitude (Mann–Whitney test, $P = 0.26$) or mean annual precipitation ($P = 0.51$).

## Sampling protocol

Our trap design was based on that in *Sydney & Gonçalves (2015)*. We cut two 2.5 cm diameter holes on opposite sides of 500-ml, disposable plastic water bottles, just below the midpoint; no landing platform was added to the traps. Each bottle trap was baited with one of three (Period I), one of five (II and III), or one of nine (IV) chemicals known to be attractive to males of a wide variety of orchid bee species. During Period I sampling, which served as a proof-of-method for the type of traps used, traps were baited with one of three aromatics: eucalyptol (1, 8-cineole), methyl salicylate, and skatole (three substances that together are known to be attractive to wide range of species (*Ackerman, 1989*)). Given our success with the traps and those baits, the Period II surveys used for the main quantitative analyses (see below) were expanded to using five baits, with the addition of eugenol and methyl vanillin (4-hydroxy-3-methoxybenzaldehyde), two others with known attractiveness (*Roubik & Hanson, 2004*). During Period III sampling, we substituted benzyl acetate (jasmine) for skatole in an attempt to capture different species; *Roubik & Hanson (2004)* rate benzyl acetate as a "good general attractant". However, during the three October surveys in 2018, we used menthol instead of eugenol because it had been shown to be especially attractive to bees of the genus *Eufriesea* in Brazil (*Dec, da Silva Mouga & Alves-dos-Santos, 2017*), and we were concerned that we had under-sampled that genus. The nine menthol-baited traps collected no orchid bees of any species, and none were

**Table 2 Sampling dates, species richness, and species diversity for Period II surveys.** Locations, sample sites, sampling dates, sample sizes, species richness, and species diversity for the Period II (2016–2017) surveys; note that when site codes are the same as those for 2015, they are not necessarily in the same location. Latilong data, trap separation distances, and habitat characteristics for each site are given in Table S2.

| Location | Location/ site code | Date sampled | Number of bees trapped | Species richness | Species diversity |
|---|---|---|---|---|---|
| Crooked Tree Wildlife Sanctuary | CT1 | 15 Dec 2016 | 33 | 3 | 1.52 |
| | CT2 | 16 Dec 2016 | 19 | 2 | 1.11 |
| | CT3 | 17 Dec 2016 | 28 | 3 | 1.44 |
| Rio Bravo Conservation Area (La Milpa) | LM1 | 19 Dec 2016 | 27 | 6 | 2.54 |
| | LM2 | 20 Dec 2016 | 40 | 6 | 3.38 |
| | LM3 | 21 Dec 2016 | 32 | 7 | 3.62 |
| | LM4 | 22 Dec 2016 | 25 | 6 | 3.51 |
| Black Rock | BR1 | 24 Dec 2016 | 35 | 10 | 3.23 |
| | BR2 | 25 Dec 2016 | 54 | 7 | 2.42 |
| | BR3 | 27 Dec 2016 | 35 | 6 | 4.39 |
| Negroman | NM1 | 28 Dec 2016 | 25 | 6 | 3.49 |
| | NM2 | 7 Jan 2017 | 19 | 6 | 4.35 |
| Las Cuevas Research Station | LC1 | 30 Dec 2016 | 7 | 2 | 1.69 |
| | LC2 | 31 Dec 2016 | 8 | 3 | 2.46 |
| | LC3 | 1 Jan 2017 | 18 | 5 | 2.89 |
| | LC4 | 2 Jan 2017 | 22 | 5 | 2.47 |
| | LC5 | 3 Jan 2017 | 8 | 4 | 2.29 |
| Chiquibul Road | CR1 | 9 Jan 2017 | 22 | 5 | 1.82 |
| | CR2 | 12 Jan 2017 | 43 | 7 | 3.88 |
| Mountain Pine Ridge | MPR | 11 Jan 2017 | 10 | 1 | 1.00 |
| Punta Gorda | PG1 | 16 Jan 2017 | 154 | 8 | 1.75 |
| | PG2 | 17 Jan 2017 | 62 | 5 | 1.88 |
| | PG3 | 25 Jan 2017 | 180 | 8 | 1.47 |
| Dump | DP | 18 Jan 2017 | 75 | 6 | 2.72 |
| Rio Blanco National Park | RB | 19 Jan 2017 | 59 | 8 | 5.87 |
| Falling Stones Butterfly Farm | FS | 22 Jan 2017 | 70 | 10 | 4.78 |
| Yemeri Grove | YG | 29 Jan 2017 | 72 | 9 | 2.89 |
| Southern Pine Savannah | PS1 | 24 Jan 2017 | 3 | 2 | 1.80 |
| | PS2 | 16 Feb 2017 | 23 | 4 | 1.87 |
| | PS3 | 17 Feb 2017 | 24 | 7 | 3.65 |
| Cockscomb Basin Wildlife Sanctuary | CS1 | 2 Feb 2017 | 55 | 6 | 2.77 |
| | CS2 | 7 Feb 2017 | 99 | 9 | 1.44 |
| | CS3 | 14 Feb 2017 | 129 | 10 | 1.86 |
| | CS4 | 19 Feb 2017 | 60 | 9 | 2.46 |
| | CS5 | 20 Feb 2017 | 62 | 9 | 2.44 |
| Hopkins area | HP1 | 3 Feb 2017 | 71 | 7 | 2.91 |
| | HP2 | 12 Feb 2017 | 33 | 9 | 2.84 |
| Hope Creek | HC1 | 11 Feb 2017 | 42 | 9 | 3.63 |
| | HC2 | 15 Feb 2017 | 25 | 7 | 3.42 |

| Location | Location/ site code | Date sampled | Number of bees trapped | Species richness | Species diversity |
|---|---|---|---|---|---|
| **Table 2** (continued) | | | | | |
| Mayflower Bocawina National Park | MB1 | 18 Feb 2017 | 47 | 11 | 5.13 |
| | MB2 | 21 Feb 2017 | 59 | 12 | 5.91 |
| | MB3 | 23 Feb 2017 | 86 | 11 | 5.12 |
| Hummingbird Highway | HH1 | 26 Feb 2017 | 22 | 7 | 3.97 |
| | HH2 | 27 Feb 2017 | 16 | 4 | 3.12 |
| Mean | | | 46.3 | 6.5 | 2.94 |
| SE | | | 5.7 | 0.4 | 0.19 |

**Table 3 Sampling dates, species richness, and species diversity for Period III surveys.** Locations, sample sites, sampling dates, sample sizes, species richness, and species diversity for the Period III (2018–2019) surveys; note that when site codes are the same as those Tables 1 and 2, they are not necessarily in the same location. Latilong data, trap separation distances, and habitat characteristics for each site are given in Table S3.

| Location | Location/site code | Date sampled | Number of bees trapped | Species richness | Species diversity |
|---|---|---|---|---|---|
| Pook's Hill | PH1 | 4 Oct 2018 | 57 | 6 | 2.66 |
| | PH2 | 4 Oct 2018 | 62 | 8 | 3.37 |
| Mayflower Bocawina National Park | MB1 | 8 Oct 2018 | 61 | 11 | 2.54 |
| | MB2 | 31 Dec 2018 | 66 | 8 | 4.51 |
| Black Rock | BR1 | 13 Dec 2018 | 77 | 7 | 2.10 |
| | BR2 | 16 Dec 2018 | 4 | 2 | 2.00 |
| Negroman | NM1 | 14 Dec 2018 | 22 | 7 | 4.25 |
| Punta Gorda | PG1 | 19 Dec 2018 | 44 | 8 | 4.59 |
| | PG2 | 20 Dec 2018 | 48 | 11 | 3.85 |
| | PG3 | 22 Mar 2019 | 83 | 10 | 5.80 |
| Falling Stones Butterfly Farm | FS | 22 Dec 2018 | 50 | 10 | 5.90 |
| Dump | DP | 25 Dec 2018 | 31 | 7 | 3.42 |
| Yemeri Grove | YG | 26 Dec 2018 | 35 | 6 | 5.26 |
| Cockscomb Basin Wildlife Sanctuary | CS | 28 Dec 2018 | 73 | 11 | 3.84 |
| Hope Creek | HC | 29 Dec 2018 | 39 | 8 | 3.06 |
| Hopkins area | HP | 30 Dec 2018 | 80 | 10 | 2.05 |
| Hummingbird Highway | HH | 27 Mar 2019 | 42 | 7 | 4.37 |
| Mean | | | 51.4 | 8.1 | 3.85 |
| SE | | | 5.3 | 0.6 | 0.30 |

attracted to baited cards during several hours of *ad lib* observations at the BR in December 2018. Therefore, we returned to use of eugenol in the remaining 14 Period III surveys. Finally, during Period IV, in an attempt to attract species absent or rare in Periods I–III, we used nine alternative baits considered to be "limited" or "good" attractants by *Roubik & Hanson (2004)*: beta-ionone, d-carvone, 1,4-dimethoxybenzene (p-dimethoxybenzene), geraniol, methyl benzoate, methyl cinnamate, p-cresol, 2-phenylethyl acetate, and a 50:50

**Table 4 Sampling dates, species richness, and species diversity for Period IV surveys.** Locations, sample sites, sampling dates, sample sizes, species richness, and species diversity for the Period IV (2019–2020) surveys; note that when site codes are the same as those Tables 1–3, they are not necessarily in the same location. Latilong data, trap separation distances, and habitat characteristics for each site are given in Table S4.

| Location | Location/site code | Date sampled | Number of bees trapped | Species richness | Species diversity |
|---|---|---|---|---|---|
| Black Rock | BR1 | 13 Dec 2019 | 11 | 4 | 3.10 |
| | BR2 | 15 Dec 2019 | 10 | 7 | 5.05 |
| Negroman | NM1 | 14 Dec 2019 | 24 | 7 | 5.56 |
| | NM2 | 15 Dec 2019 | 38 | 6 | 4.88 |
| Punta Gorda | PG1 | 17 Dec 2019 | 117 | 11 | 4.95 |
| | PG2 | 18 Dec 2019 | 139 | 10 | 3.52 |
| Yemeri Grove | YG | 21 Dec 2019 | 12 | 4 | 3.13 |
| Dump | DP | 21 Dec 2019 | 51 | 8 | 1.93 |
| Hopkins area | HP | 23 Dec 2019 | 17 | 5 | 3.04 |
| Cockscomb Basin Wildlife Sanctuary | CS1 | 25 Dec 2019 | 110 | 5 | 3.48 |
| | CS2 | 27 Dec 2019 | 131 | 6 | 3.59 |
| | CS3 | 28 Dec 2019 | 39 | 6 | 3.37 |
| | CS4 | 30 Dec 2019 | 127 | 6 | 2.87 |
| | CS5 | 1 Jan 2020 | 11 | 4 | 3.67 |
| Hope Creek | HC | 2 Jan 2020 | 65 | 11 | 6.80 |
| Mean | | | 60.1 | 6.7 | 3.93 |
| SE | | | 13.0 | 0.6 | 0.26 |

mix (by volume) of eugenol and methyl salicylate (We will present data on the association of orchid bee species with different baits in a separate publication). Altering the set of baits used from one sampling period to the next did not affect our analyses, because "period" was not used as a factor in any of our statistical tests, though we do note the effects of changing baits in the discussion.

All chemicals were purchased as liquids or solids from Consolidated Chemicals and Solvents (Quakertown, Pennsylvania, USA), Sigma Aldrich (St. Louis, Missouri, USA), or HiMedia Laboratories (Mumbai, India). All liquids all were pure solutions and used in undiluted form. For the baits purchased as solids (1, 4-dimethoxybenzene, methyl cinnamate, and methyl vanillin), we prepared saturated solutions in 40% ethanol. Because the 40% ethanol was present in all traps as a killing agent, its use for these baits did not add an extra potential attractant.

To bait each trap, we bound together two cotton swabs with wire, soaked one end of the pair in a single chemical solution, and suspended them baited-ends downwards from inside the top of the bottle, the cap holding the wire in place. When deployed in the field, we added 40% ethanol (vodka) to each bottle to a depth of ~3 cm to serve as a killing agent and preservative. On clear or partly-cloudy days, we placed traps in the field from 08:30–09:30 h and retrieved them from 14:30–15:30 h. The cotton-swabs and killing agent were never reused in subsequent samples. On each of 10 sampling days in Period I (Table 1), we placed a total of 12 traps comprised of four sets of three traps, each set

containing three bottles with one each of the three baits (720 trap-hours). On each of 44 days during Period II and 17 days in Period III (Tables 2 and 3), we placed 15 total traps comprised of three sets of five traps, with each set containing one each of five baits (5,490 trap-hours). For Period IV, because of the greater number of baits, each of the 15 transects included two sets of the nine baits (1,620 trap-hours). For each sample, the order of the baits within a set was randomized, and the same order was repeated within each set on the same day.

We positioned transects along roads or trails, or at the edges of clearings, the bottles suspended from vegetation at heights of 1.5–2.0 m. During surveys in Brazil, *Ribeiro et al. (2022)* found that abundance of orchid bee species was highest in forest understory traps at 1.5 m height compared to the canopy (at 12 m in their study) and that species richness did not differ between the understory and canopy; all 13 species they did not trap in the understory or which were represented as singletons were also rare in the canopy. For each transect, we attempted to place each set of traps within similar habitats, with the result that the distances between sets varied among sites (Tables S1–S4). Within locations (Fig. 1) during any period, transects on different days never overlapped spatially, so we treated them as separate sites. When samples were retrieved each day, bees from each chemical bait were combined in Whirl-Pak® bags along with the ethanol from the trap.

For logistic reasons, we were unable to randomize the dates on which we sampled across the entire country. As a result, there were significant correlations in 2016–2017 between the date on which samples were taken and both latitude (Spearman rank correlation; $r_S = -0.38$, $P < 0.05$) and mean annual precipitation ($r_S = 0.68$, $P < 0.0001$), though not elevation ($r_S = -0.26$, $P = 0.09$). However, the (decimal) latitudinal difference between the first and last sample taken during the extensive Period II survey was <0.7°, and the elevational difference was just 60 m.

Bees were identified using *Kimsey (1979, 1982)*, *Roubik & Hanson (2004)*, *Roubik (2006)*, and *Eltz et al. (2011)*. To identify orchid bees, we used a stepwise process. Two of us (KMO and CMD) began by independently identifying the bees using microscopes. We then compared our results, which almost always coincided. Any disagreements were discussed until we came to a consensus, and we never ended by agreeing to disagree for any specimens. In relatively difficult cases, we used authoritatively identified reference specimens in the Cornell University Insect Collection, the Invertebrate Zoology Collections of the American Museum of Natural History, and the USDA-ARS Pollinating Insect-Biology, Management, Systematics Research collection, Logan, UT. Voucher specimens are deposited in the Montana Entomology Collection at Montana State University—Bozeman. Bees were collected with permission from the Belize Forestry Department; Permit Reference Numbers: CD/60/3/15(43), WL/1/1/16(52), WL/2/1/18 (38), and FD/WL/1/19(33) issued to KMO and RPO and CD/60/3/15(25), WL/2/1/17(05), and WL/2/1/18(32) issued to CMD and JBR.

## Data analysis

All means are presented ± standard errors, with the exception of Tables S5–S8, where standard deviations (SD) are presented so that coefficients of variation (CV = mean/SD)

could be used to compare among-sample variation in numbers of individuals for each species within survey periods. Species diversity was characterized using the Hill's #2 Index ($1/\lambda$), the inverse of Simpson's Diversity Index, ($\lambda = \Sigma p^2$), where p is the proportional abundance of each species in a sample (*Ludwig & Reynolds, 1988*); the index ranges from 1.0 upwards (where a sample with a single species has an index value of 1.0). Unless otherwise stated, correlation coefficients are from Spearman rank correlation analyses conducted using SigmaPlot, v. 11 (*Systat Software, Inc, 2001*).

We used EstimateS (*Colwell, 2013*) to generate a Chao1 species richness predictor using all 86 samples (arranged from first to last date of sampling) from all four periods to estimate the number of bee species present at the sites we sampled. Though Chao1 is a good estimator of minimum species richness, it often underestimates the true parameter (*Chao, 1984*; *Gotelli & Colwell, 2011*). Each sample was randomized 100 times to create the mean species accumulation curve. We also generated a species accumulation curve with raw data for comparison to the Chao1 estimate.

We focused on the Period II data to examine how species richness, species diversity, and relative abundance of species varied among sites and habitats. We restrict the analyses to Period II, because it was the largest (44 of the 86 samples) and most geographically wide-ranging (18 locations compared 7–12 in the other periods). Including the other survey periods in the analyses would have introduced variation due to different baits, numbers of traps per transect, and numbers of transects per sample. We used Kruskal–Wallis tests (with Dunn's comparisons; SigmaPlot v. 11, Systat Software, Inc., Chicago, IL, USA) to test whether species richness and species diversity varied among three general habitat classes (Tables S1–S4): (1) sites with agricultural presence ($N = 13$), (2) pine forests and pine savannah, none of which had agricultural presence ($N = 6$), and (3) those broadleaf forests lacking agricultural presence ($N = 25$). For six common species, we used $2 \times 2$ chi-square contingency table analyses (*Lowry, 2022*) to examine whether the proportion of each species (relative to that of all other species) differed between sites with and without agricultural presence (all samples combined for each category).

We conducted a canonical correspondence analysis (CCA; Vegan package in R, version 3.4.1; *R Core Team, 2017*) based on a sample-by-species matrix, and a corresponding sample-by-environmental variable matrix, which included information of the latitude, elevation, and historical mean annual precipitation of each site. The CCA analysis allowed us to visualize which sites and species were associated with gradients of each environmental variable. While these analyses were restricted to Period II data, we note in the discussion where data from the other periods is congruent or dissimilar to those results.

## RESULTS

### Overview of species collected

The 1,275 baited-traps in the four survey periods captured 4,571 male orchid bees, and no females. Among the 24 species collected, *Euglossa* Latreille was the most species-rich genus (16 species; 77.7% of all orchid bee individuals), followed by *Eulaema* Lepeletier (three species; 18.3%), *Eufriesea* Cockerell (three species; 1.0%), and the genus of brood-parasites *Exaerete* Hoffmansegg (two species; 2.9%) (Table 5).

**Table 5 Mean (±SE) number of bees of each species present in samples during each sampling period.** Species abbreviations are those used in Figs. 1 and 4. Note that comparisons among survey periods are confounded by number and types of baits used (see Materials and Methods and Discussion).

| Species | Sampling period (years; number of samples) | | | |
| --- | --- | --- | --- | --- |
| | Period I (2015; $n = 10$) | Period II (2016–2017; $n = 44$) | Period III (2018–2019; $n = 17$) | Period IV (2019–2020; $n = 15$) |
| *Eufriesea concava* Friese (Efcon) | 0.50 ± 0.40 | 0.32 ± 0.14 | 0.47 ± 0.41 | – |
| *Eufriesea rugosa* Friese (Efrug) | 0.60 ± 0.60 | 0.23 ± 0.14 | 0.18 ± 0.18 | – |
| *Eufriesea schmidtiana* Friese (Efsch) | – | – | 0.06 ± 0.06 | – |
| *Euglossa allosticta* Moure (Egall) | 1.30 ± 042 | 0.27 ± 0.09 | 0.24 ± 0.18 | – |
| *Euglossa bursigera* Moure (Egbur) | – | – | 0.06 ± 0.06 | – |
| *Euglossa cyanura* Cockerell (Egcya) | – | – | – | 0.07 ± 0.07 |
| *Euglossa deceptrix* Moure (Egdec) | – | 0.30 ± 0.20 | 0.59 ± 0.26 | 0.13 ± 0.09 |
| *Euglossa dilemma* Bembé and Eltz (Egdil) | – | 0.14 ± 0.07 | 0.24 ± 0.18 | 1.07 ± 0.40 |
| *Euglossa hansoni* Moure (Eghan) | 0.10 ± 0.10 | – | – | – |
| *Euglossa hemichlora* Cockerell (Eghem) | – | – | – | 10.40 ± 3.64 |
| *Euglossa heterosticta* Moure (Eghet) | 5.00 ± 1.27 | 1.05 ± 0.33 | 3.88 ± 1.69 | 0.53 ± 0.38 |
| *Euglossa ignita* Smith (Egign) | 9.70 ± 4.79 | 0.93 ± 0.36 | 2.59 ± 0.78 | 0.13 ± 0.00 |
| *Euglossa imperialis* Cockerell (Egimp) | 8.80 ± 4.35 | 3.18 ± 0.78 | 9.35 ±3.32 | – |
| *Euglossa mixta* Friese (Egmix) | 9.30 ± 2.49 | 1.73 ± 0.32 | 2.71 ± 0.93 | 14.07 ± 4.78 |
| *Euglossa obtusa* Dressler (Egobt) | 1.80 ± 0.70 | 0.82 ± 0.24 | 0.88 ± 0.26 | 3.73 ± 1.61 |
| *Euglossa purpurea* Friese (Egpur) | 21.20 ± 8.77 | 17.59 ± 4.79 | – | 8.60 ± 3.40 |
| *Euglossa tridentata* Moure (Egtri) | 5.20 ± 1.13 | 3.89 ± 1.00 | 14.71 ± 2.96 | 7.40 ± 2.34 |
| *Euglossa variabilis* Friese (Egvar) | 0.20 ± 0.13 | 0.41 ± 0.18 | 0.77 ± 0.59 | 0.07 ± 0.07 |
| *Euglossa viridissima* Friese (Egvir) | 0.40 ± 0.40 | 2.34 ± 0.90 | 1.41 ± 0.56 | 10.53 ± 4.74 |
| *Eulaema cingulata* (F.) (Elmar) | 9.60 ± 2.30 | 11.07 ± 1.02 | 6.65 ± 1.61 | 0.27 ± 0.15 |
| *Eulaema meriana* (Olivier) (Elmer) | 0.20 ± 0.13 | 0.59 ± 0.33 | 3.35 ± 1.71 | 0.07 ± 0.07 |
| *Eulaema polychroma* (Mocsáry) (Elpol) | – | 0.18 ± 0.07 | 0.24 ± 0.18 | 2.67 ± 0.81 |
| *Exaerete frontalis* (Guérin-Méneville) (Exfro) | 0.10 ± 0.10 | 0.32 ± 0.13 | 0.35 ± 0.15 | – |
| *Exaerete smaragdina* (Guérin-Méneville) (Exsma) | 1.70 ± 0.82 | 0.98 ± 0.22 | 2.71 ± 1.12 | 0.40 ± 0.24 |

Within each survey period, we observed considerable differences in abundances of species, as well as among-sample variation in counts of each species, as indicated by CV values (Tables 1–4, S5–S8). In Period I, five of 17 species accounted for >75% of orchid bees captured: *Eg. purpurea, Eg. ignita, El. cingulata, Eg. mixta,* and *Eg. imperialis* (Table S5). The lowest CV values were for *Eg. heterosticta, Eg. mixta, Eg. tridentata,* and *El. cingulata,* the latter two being the only species present in all 10 samples. In contrast, seven species each made up <1% of the total 2015 catch and were found in just 1–2 samples, and so had high CV values; *Eg. hansoni* and *Ex. frontalis* were represented by one specimen each.

In Period II (Tables 2 and S6), four of the 19 species collected accounted for >75% of the 2,038 bees trapped: *Eg. purpurea, El. cingulata, Eg. tridentata,* and *Eg. imperialis.* Eight

other species each comprised <1% of the total catch. The five species with the lowest CVs (*i.e.*, more common) were *Eg. imperialis*, *Eg. mixta*, *Eg. tridentata*, *El. cingulata*, and *Ex. smaragdina*, whereas those with the highest CVs were the rarer *Ef. concava, Ef. rugosa, Eg. deceptrix, Eug dilemma, Eg. variabilis*, and *El. meriana*. The three species collected in Period II, but not found in Period I were *Eg. dilemma* (at HP), *Eg. deceptrix* (at NM), and *El. polychroma* (at BR).

In Period III (Tables 3 and S7), four of 20 species comprised two-thirds of the 874 bees captured: *Eg. tridentata, Eg. imperialis, El. cingulata*, and *Eg. heterosticta*. *Euglossa tridentata* and *Eg. cingulata* had the lowest CV values, whereas *Ef. concava, Ef. rugosa, Ef. schmidtiana, Eg. bursigera, Eg. dilemma*, and *El. polychroma* had the highest, each appearing in just one or two samples. *Eufriesea schmidtiana* and *Eg. bursigera* appeared as singletons, and were the only two species not trapped in Periods I or II. One major change was the absence *Eg. purpurea* in Period III samples, the result of not using skatole-baited traps in Period III, which had captured 99.8% of *Eg. purpurea* in Periods I and II.

Results from Period IV (Tables 4 and S8) differed most from Periods I–III because of the use of the completely different set of baits; we collected 902 bees of 16 species, the most frequent being *Eg. mixta, Eg. viridissima, Eg. hemichlora, Eg. purpurea*, and *Eg. tridentata*. The previously uncommon *El. polychroma* was attracted in relatively higher numbers (40 bees, across eight samples, all in traps baited with beta-ionone). The greatest difference from the previous surveys was the presence of *Eg. hemichlora* in 12 samples, mostly in the Stann Creek District and all collected in 1,4-dimethoxybenzene-baited traps. Among the singletons, were *Eg. cyanura* (not collected previously by us), *Eg. variabilis*, and *El. meriana*. Several species attracted mainly to cineole in previous periods were rare (*El. cingulata*) or absent (*Ef. concava, Ef. rugosa, Eg. allosticta*, and *Eg. imperialis*) in traps in Period IV.

## Patterns in species richness, diversity, and abundance

The 86 samples throughout Belize garnered 1–12 species each (Tables 1–4, S5–S8). Among the eight samples with 1–3 species, five (all in Period II) were taken in relatively open habitats at sites with greater spacing among trees than most of the broadleaf forests sampled (*i.e.*, CT1-3, MPR, and PS1). At LC, a dense submontane forest and the location with the highest elevation, low specimen and species counts were likely related to cool weather during sampling (Table S2). In contrast to these species-poor samples, all 15 samples with ≥10 species were from densely-treed habitats (Tables 1–4, S1–S4). Such high species-counts were especially common at CS (three samples), PG (6), and MB (4).

Among the 44 Period II samples, species richness was correlated with historical mean annual precipitation ($r = 0.40$, $P = 0.007$, $n = 44$), but not with latitude ($r = -0.18$, $P = 0.25$) or elevation ($r = -0.15$, $P = 0.32$). This was the case even though precipitation was correlated with both latitude ($r = -0.76$, $P = 0.0000002$) and elevation ($r = -0.52$, $P = 0.0003$). Elevation and latitude were uncorrelated ($r = 0.16$, $P = 0.30$). Species richness tended to be higher in samples with greater numbers of bees, but the number of bees per sample explained only about a third of the variation in species richness (linear regression, $r^2 = 0.32$, $F_{1, 42} = 19.81$, $P < 0.001$).

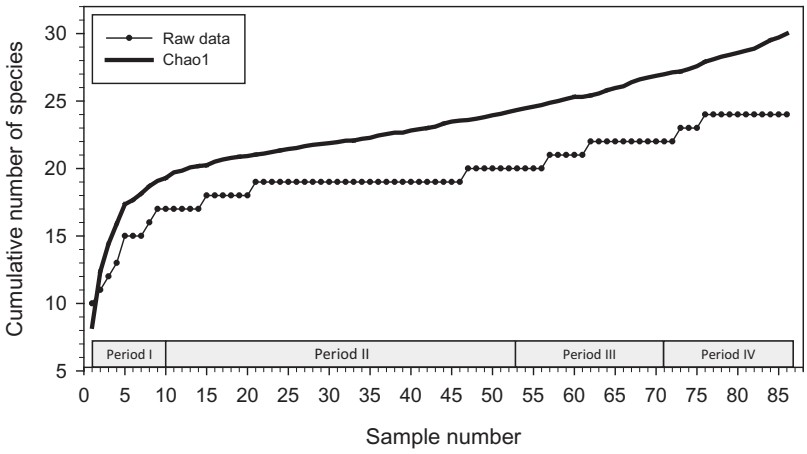

**Figure 2 Cumulative number of species added to species list during study.** Species accumulation curve (with raw data; lower line) and Chao1 mean (upper line) for orchid bees sampled in 2015–2020 (86 samples).

In Period II, the species diversity index was not correlated with latitude (Spearman rank correlation, $r = 0.12$, $P = 0.42$), elevation ($r = 0.26$, $P = 0.09$), precipitation ($r = 0.05$, $P = 0.75$), or specimen number per sample ($r = 0.16$, $P = 0.18$). Abundance (total number of bees per sample transect) was positively correlated with historical precipitation ($r = 0.51$, $P < 0.001$), but negatively correlated with both latitude ($r = -0.33$, $P = 0.03$) and elevation ($r = -0.41$, $P = 0.006$). Abundance did not vary between sites with (mean = 38.7 ± 5.8 bees) and without (mean = 49.5 ± 7.7 bees) agricultural presence (Mann–Whitney test, $P = 0.77$).

Analysis with EstimateS yielded a Chao1 mean prediction of 30 orchid bee species present at the locations we sampled from December 2015 to January 2020 (SD = 7.27; 95% confidence interval of 24.9–62.7 species) (Fig. 2). There was considerable uncertainly around that estimate, and no indication that an asymptote had been reached during sampling.

The positions of the sites and the direction of the gradients on the CCA site-plot for Period II (Fig. 3) reveals several clear biogeographic patterns. First, the precipitation gradient runs roughly opposite to the elevational and latitudinal gradients because wetter sites were generally further south and east in the lowlands of the Toledo and Stann Creek Districts (Fig. 1).

Second, when multiple samples were taken on different days at sites at the same general locations (*i.e.*, BR, CS, CT, HC, HH, HP, LC, LM, MB, NM, and PG), they tended to cluster near one another on the plot, indicating that traps caught similar arrays of species when sites were near one another in the same habitat.

Third, the three sites at CT, the only location with samples dominated by *Eg. viridissima* in Period II (88% of all bees trapped and just 1–2 other species per sample), are clustered together in the lower left of the plot, well-separated from other samples. CT lies at 15 m elevation and had the only sites classified as lowland swamp forest or lowland dense forest (Table S2).

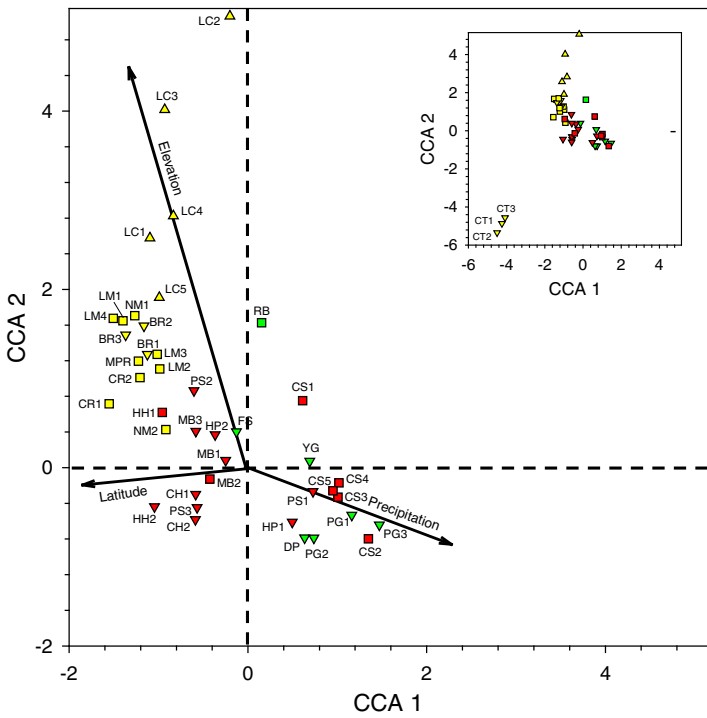

**Figure 3 Canonical correspondence analysis plot of 44 sites sampled during 2016–2017 survey period.** See Table 2 for site abbreviations. Symbol shapes refer to elevation classes arbitrarily chosen for illustration purposes: 580–600 m (upward pointing triangles), 60–240 m (squares), and <60 m (downward pointing triangles) (see Table S2 for latilong, elevation, and precipitation values for each site). Symbol colors refer to three precipitation classes arbitrarily chosen for illustration purposes: relatively low (yellow), intermediate (green), and relatively high (red) precipitation. The inset shows all sites in the larger figure along with the three CT sites in the lower left.

Fourth, the samples in the lower right quadrant of the CCA plot, as well as others near the zero-intercept, are from lower latitude and low- to mid-elevation sites (15–120 m). These include most Toledo District sites (DP, FS, PG, and YG) and all Stann Creek District sites (CS, HC, HP, MB, and PS). The most common species in 21 samples from these sites were (1) *Eg. purpurea* which was present in all samples of those districts and which comprised 53.4% of 1,490 orchid bees collected and), (2) *El. cingulata* (17.1%, present in all samples), and (3) *Eg. tridentata* (8.9%, present in 17 samples) (Tables 2 and S5). *Euglossa viridissima*, in contrast, comprised just 1.7% of bees collected those sites; it was, however, common at these same sites when different baits were used in Period IV, so may be more widely distributed than the results from Period II indicate. Toledo/Stann Creek samples include several species rare to absent from samples farther north and west including: *Eg. allosticta, Eg. heterosticta, Eg. ignita, El. meriana,* and *Ex. frontalis* (Table S5). A more southerly distribution of these five species was also observed in Period I and III samples (Tables S5 and S7), and for *Eg. ignita* in Period IV (Table S8).

Finally, whereas the Toledo/Stann Creek sites are at the relatively mesic end of a precipitation gradient, the opposite end of that gradient includes a group of drier, higher elevation (85–600 m), mid- to higher-latitude sites in the Cayo and Orange Walk Districts (BR, CR, LC, LM, NM, and MPR). The three most common species among the 510 bees

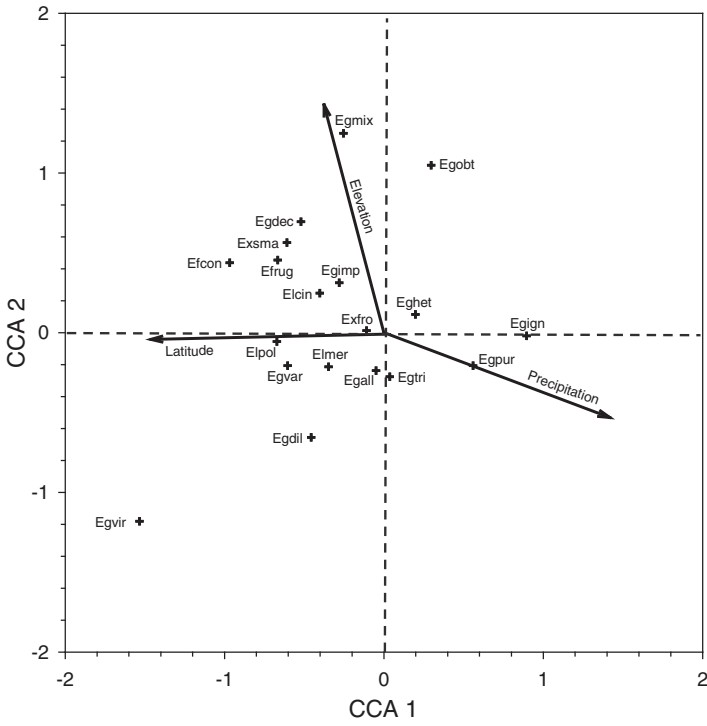

**Figure 4 Canonical correspondence analysis plot for species collected during Period II.** Canonical correspondence analysis plot of 19 species of orchid bees collected in Period II. See Table 5 for species abbreviations.                

collected at these sites in Period II were *El. cingulata* (48.8% of bees, present in all samples), *Eg. imperialis* (14.4%, present at 13 of 17 sites), and *Eg. mixta* (8.8%, 11 of 17 sites) (Table S5). *Euglossa purpurea*, by far the most common bee in Toledo/Stann Creek, was extremely rare (0.2% of bees) at Cayo/Orange Walk sites (Table S5). The Cayo/Orange Walk samples included three species, *Ef. concava*, *Ef. rugosa*, and *Eg. deceptrix* absent from Toledo District and Stann Creek District samples (though a single *Eg. deceptrix* was later captured at PG-3 in Period III) (Table S6).

Inspection of the CCA species-plot (Fig. 4) reinforces the observed site patterns. *Euglossa purpurea* is a common species that lies at the end of the elevation (low), latitude (low), and precipitation (wet) gradients. The mean elevation of sites where *Eg. purpurea* was present (mean = 79.4 ± 24.4 m) was lower than that for *Eg. mixta* (mean = 166.9 ± 37.7 m; Mann–Whitney Test, $P = 0.034$), which lies at the higher elevation end of the gradient (though *Eg. mixta* was present in many low elevation sites). The same is true for latitude, as the mean decimal latitude of sites where *Eg. purpurea* was present (16.65 ± 0.069°N) was lower than where *Eg. viridissima* occurred (16.96 ± 0.12°N; Mann–Whitney Test, $P = 0.04$), a mean equivalent of ~35 km of latitude. Not surprisingly, because of the correlations between latitude and precipitation in Belize, the mean annual precipitation values for sites where *Eg. purpurea* was present (2,557 ± 105 mm, $n = 24$) were higher than for sites where either *Eg. cingulata* (2,173 ± 88 mm, $n = 44$; Mann–Whitney test, $P = 0.002$) or *Eg. imperialis* occurred (2,126 ± 136 mm; $n = 24$; Mann–Whitney Test, $P = 0.003$).

**Table 6 Species richness and diversity of orchid bees in three general habitat classes.** Species richness and diversity in three general habitat classes in Belize (2016–2017 data). Numbers followed by the same letters are not significantly different from one another; those with different letters are significantly different (Dunn's test, $P < 0.05$).

| Habitat classes | Number of samples | Species richness | Species diversity |
|---|---|---|---|
| Sites with agricultural presence | 13 | 7.00 ± 0.54a | 3.60 ± 0.26a |
| Pine forest and pine savannah | 6 | 4.17 ± 1.30a | 2.18 ± 0.48b |
| Broadleaf forests | 25 | 6.84 ± 0.54a | 2.77 ± 0.25b |
| Kruskal-Wallis test (d.f. = 2) | | H = 9.79 P = 0.083 | H = 4.98 P = 0.008 |

**Table 7 Proportions of common species at sites without and with integrated or adjacent agricultural activities.** Proportions of six common species within combined samples at sites without (0) and with (1) integrated or adjacent agricultural activities (2016–2017 survey period). Chi-square contingency table comparisons for each species are between sites with 0 and 1 index values, comparing numbers for each species listed in table with all other species combined. d.f. = 1 for all tests.

| Species | Agricultural presence index (number of samples) | | Chi-square contingency table analysis | |
|---|---|---|---|---|
| | 0 ($n = 31$) | 1 ($n = 13$) | $\chi^2$ | $P$ |
| Eg. heterosticta | 0.027 | 0.010 | 4.10 | 0.042 |
| Eg. imperialis | 0.061 | 0.093 | 5.89 | 0.015 |
| Eg. mixta | 0.044 | 0.018 | 6.30 | 0.012 |
| Eg. purpurea | 0.442 | 0.189 | 102.27 | <0.0001 |
| Eg. tridentata | 0.043 | 0.209 | 133.26 | <0.0001 |
| El. cingulata | 0.220 | 0.296 | 11.63 | <0.001 |

Species richness did not vary among the three general classes of habitats we established in Period II, but species diversity was highest on average at sites with agricultural presence (Table 6). Comparing the distribution of some of the most common species, *Eg. imperialis*, *Eg. tridentata*, and *Eul cingulata* made up a greater proportion of samples at sites with agricultural presence, whereas *Eg. heterosticta*, *Eg. mixta*, and *Eg. purpurea* were more common at sites without agricultural presence (Table 7).

## DISCUSSION

### Distribution of Euglossini within Belize

Our data suggest a considerable disparity in distribution of orchid bee species in Belize among locations we sampled during the late-wet and early-dry seasons. Based on combined records across the survey periods, species such as *Eg. imperialis, Eg. mixta, Eg. tridentata, Eg. viridissima, El. cingulata*, and *Ex. smaragdina* were widely distributed. Those with apparently more restricted distribution include (1) *Ef. concava* and *Ef. rugosa*, found by us only in the northwest and west-central locations at LM, BR, NM, and CR, (2)

*Eg. ignita* and *Eg. purpurea*, mainly in southern Belize, (3) *El. meriana*, most abundant at MB sites in Stann Creek, and (4) *Ex. frontalis*, found mainly in the Stann Creek District. The overall differences in the distribution and relative abundances of the most common species (as well as *Eufriesea* spp.) collected in Period II surveys can be seen in the in Fig. 1. The sites and taxa presented in the figure encompass 80% of the bees collected during that period, though the histograms leave out a few locally abundant species such as *Eg. variabilis* at HH and *El. meriana* at MB.

We provide a preliminary analysis of how the distribution of orchid bees relates to latitude, elevation, and mean precipitation. For this, we use mainly the data from Period II, with several notes on data from other periods (without quantitative comparisons). Overall, species richness (but not diversity) was positively correlated with mean annual historical precipitation, but neither richness nor species diversity were correlated with latitude and elevation (despite the correlation of both variables with precipitation). However, sites with comparable latitudes in the Period II samples (Fig. 1, Table S2) commonly harbored different species assemblages. The sites at CT1-3 and LM1-4 in the north were all within 0.07° latitude and were sampled within 8 days of one another; CT was about 145 m lower in elevation and is slightly drier on average, and the two areas are within different vegetation zones, as classified by *Meerman & Sabido (2001)*. CT and LM cluster at different positions on the CCA plot because CT samples were dominated by *Eg. viridissima* (88% of bees), whereas at the more highly forested LM sites, most bees captured were either *El. cingulata* or *Eg. imperialis* (and *Eg. viridissima* was rare).

Similarly, two sets of Stann Creek District sites (MB1-3 and CS1-5) were within ~0.17° latitude and sampled within three weeks of one another; MB and CS are both classified as "tropical evergreen seasonal broadleaf lowland forest over poor or sandy soils" (*Meerman & Sabido, 2001*). However, at the CS sites *Eg. purpurea* and *El. cingulata* comprised >80% of specimens in Period II, whereas *Eg. imperialis, Eg., purpurea, Eg. tridentata, El. cingulata*, and *El. meriana* were all relatively common at MB (in three samples); MB was the only location in Belize where *El. meriana* was relatively common in Period II (as well as in the two Period III samples).

Lastly, in the far south, the RB, DP, and the PG1-3 sites spanned <0.15° latitude and were sampled within 10 days of one another in Period II. RB was >200 m higher in elevation than the other two locations and was historically the driest, and whereas DP was a marshy, partly-abandoned rice field, both RB and DP were densely forested (Table S2). At RB, five species were relatively common, and none comprised more the 28% of the sample (that being *Eg. imperialis*, which was rare or absent in the DP and PG samples). In the DP sample, *Eg. tridentata* (52%) was predominant (as it was in all four survey periods), whereas *Eg. purpurea* (71–82%) was the most abundant orchid bee in all three PG samples in Period II (but not in Period III when skatole was not used in traps).

The only other quantitative survey of orchid bees in Belize that we are aware of was that *Schüepp, Rittner & Entling (2012)*, conducted in the northeast Corozal District using Malaise traps baited with a chemical blend of cineole, eugenol, and methyl salicylate rather than single baits. Their traps collected mainly *Eg. viridissima* (>96% of orchid bees), mirroring the predominance of *Eg. viridissima* in our samples at CT, the closest of our

sites. Thus, we initially concluded that *Eg. viridissima* is most common in relatively drier, northern Belize. However, in Period IV it was relatively common in PG samples in the south when it was attracted to the dimethoxybenzene-baited traps not used previously.

Several studies have addressed the effects of habitat disturbance, sometimes caused by agriculture, on the abundance and distribution of orchid bees in the Neotropics (*e.g.*, *Otero & Sandino, 2003*; *Brosi et al., 2009*; *Schüepp, Rittiner & Entling, 2012*; *Livingston et al., 2013*; *Botsch et al., 2017*; *Cândido et al., 2018*; *Allen et al., 2019*; *Opedal, Martins & Marjakangas, 2020*; *Roubik et al., 2021*). *Otero & Sandino (2003)*, for example, found species differences in abundance among sites in Columbia, with some species (*e.g.*, *El. cingulata*) being more frequent on farms, whereas others (including *Eg. imperialis*) were more abundant in secondary forests.

In our study, the distribution of euglossines was also non-random relative to the presence of agricultural activities, which we assume to be a general indicator of habitat fragmentation due to human activities. *Euglossa imperialis*, *Eg. tridentata*, and *El. cingulata* were more common at sites with agricultural presence within or adjacent to the areas sampled. However, *Eg. heterosticta*, *Eg. mixta*, and *Eg. purpurea* tended to be present in relatively higher numbers at sites we recorded as having no agricultural presence. Because of the relatively coarse way in which we categorized sites, our conclusions on the effects of agricultural activities on orchid bees are clearly preliminary, but along with those of *Schüepp, Rittiner & Entling (2012)*, they suggest a fruitful avenue for future research on Belizean orchid bees. The need to further study the effect of land-use changes on bees was demonstrated by *De Palma et al. (2016)* who showed that the effects of agriculture on overall bee diversity can be region-specific and argued that studies are needed outside of well-studied taxa such as bumble bees and of the ecosystems of Europe and North America.

## Species of Euglossini documented within Belize

Our sampling with baited traps in Belize from 2015–2020 garnered 24 species of orchid bees, to which we can add *Eulaema luteola* Moure, based on a single female we netted on a flower at PH (Cayo District) in March 2019. *Ascher & Pickering (2019)* indicate that *Euglossa townsendi* Cockerell) and *Exaerete dentata* (L.) have been collected in Belize, bringing the known total to 27 species. Most of our 25 species records do not represent northwards latitudinal range expansions from previous studies. Sixteen of the species are reported from Mexico (*Ayala, Griswold & Yanega, 1996*; *Eltz et al., 2007*), and *Ef. schmidtiana* (a singleton in our data) is recorded from Belize in *Moure & Melo (2022)*. Among the other species we collected as singletons, *Eg. bursigera* and *Eg. cyanura* have been collected in the Cayo District of Belize (*Ascher & Pickering, 2019*), while *Eg. hansoni* is reported from Honduras (*Moure & Melo, 2022*; *McCravy et al., 2016*; *Ascher & Pickering, 2019*) at about the same latitude as the southern border of Belize. We collected both *Eg. bursigera* and *Eg. cyanura* at our southernmost location (PG).

Thus, with the *Eg. townsendi* and *Ex. dentata* records, at least 27 Euglossini are documented for Belize, seven of which apparently do not occur south of Central America: *Ef. rugosa*, *Eg. cyanura*, *Eg. dilemma*, *Eg. obtusa*, *Eg. purpurea*, *Eg. viridissima*, and

**Table 8 Numbers of orchid bee species in Belize and other countries of region.** Number of documented orchid bee species occurring in countries of Central America and Mexico, and the number of species per unit land area.

| Country | Number of species | Land area (km²)[*] | Species / 1,000 km² | References[**] |
|---|---|---|---|---|
| Belize | 27 | 22,966 | 1.18 | Present study; *Moure & Melo (2022)* |
| Costa Rica | 65 | 51,100 | 1.27 | *Roubik & Hanson (2004)* |
| El Salvador | 5 | 21,040 | 0.24 | – |
| Honduras | 34 | 112,090 | 0.30 | *McCravy et al. (2016)* |
| Mexico | 36 | 758,400 | 0.05 | *Ayala, Griswold & Yanega (1996)* |
| Nicaragua | 33 | 130,373 | 0.25 | *Hinojosa-Díaz & Engel (2012)* |
| Panama | 69 | 78,200 | 0.88 | *Roubik & Hanson (2004)* |

Notes:
[*] https://simple.wikipedia.org/wiki/List_of_countries_by_area.
[**] Supplemented by fully-documented records in *Ascher & Pickering (2019)*.

*El. cingulata* (*Roubik & Hanson, 2004*; *Skov & Wiley, 2005*; *Eltz et al., 2011*; *Moure & Melo, 2022*; *Ascher & Pickering, 2019*). Although Belize has fewer documented species than most other countries in the region (Table 8), only Costa Rica has a higher number relative to land area, perhaps because Belize has the highest percentage of forested land in Central America (*Corrales, Bouroncle & Zamora, 2015*) and the lowest human population density (*World Population Review, 2018*). However, the lower values for some countries are undoubtedly due partly to lower sampling intensity, and our own analysis does not provide evidence that habitat fragmentation due to agriculture reduces species diversity.

Our Chao1 analysis provided no suggestion that we reached an asymptote in the cumulative number of species collected at locations we visited and with the baits that we used. Nineteen of the 24 species we collected in traps were detected by the twenty-first sample (of 86 total), and another 25 samples were needed to add the twentieth species. The addition of new species was often related to our switch to using alternative baits or to repeated sampling at the same locations. When we added vanillin and eugenol for Period II, the first three previously uncollected species were attracted with those baits (*El. polychroma*, *Eg. deceptrix*, and *Eg. dilemma*). Similarly, the final two species (*Eg. hemichlora* and *Eg. cyanura*) were found at the beginning Period IV, both attracted to baits not used previously. It is also clear that repeated sampling at the same sites, even at similar times of year with similar baits, is required to collected uncommon species, as several examples illustrate. *Euglossa bursigera* was not collected until the eighth transect at the PG location (by which time 726 bees had been collected at that location) and *Ef. schmidtiana* was not found until the fourth sample at Mayflower Bocawina (when 253 bees had been collected at that location). Both were attracted to cineole, which we had used since the beginning of the study.

Thus, it remains to be seen how many species of Euglossini occur in Belize. Information on the geographic occurrence of other euglossines whose distribution matches or straddles the latitudes of Belize suggests that 19 other species (Table S9) might also be in the country, assuming that suitable habitats and floral resources are present, and that colonization

opportunities have existed. If our list of 24 species from the traps underestimates the number of species in Belize, we can conceive of several possible reasons. First, in many of our surveys, we used a small subset of the dozens of baits that have been used to attract orchid bees (*Roubik & Hanson, 2004*). However, we did use bait and geographic range information summarized in *Roubik & Hanson (2004)* to devise a set of baits known to be most attractive to the species that occur in northern Central America. Furthermore, when *Ackerman (1989)* used 16 different baits during extensive year-long sampling in Panama, 51 of 52 species he observed were attracted to at least one of the baits that we used on most of our transects (*i.e.*, cineole, eugenol, methyl salicylate, skatole, and vanillin). Nevertheless, the fact that we added two species to our records when we switched to nine alternative baits in Period IV suggests the need to sample more widely with an expanded set of baits.

A second concern is that the efficiency of traps in capturing bees varies among species and genera, and that traps provide lower estimates of species richness than when bees are individually netted at baited stations (*Mattozo, Faria & Melo, 2011*; *Nemésio & Vasconcelos, 2014*). The fact that we collected just one individual each of *Eg. bursigera*, *Eg. cyanura*, *Eg. hansoni* and *Ef. schmidtiana*, could reflect reluctance of their males to enter traps. It is also possible that, when one species is attracted in very high numbers to traps, the intense flurry of activity around the small entry holes repels some species. However, these potential limitations of traps have to be balanced against the fact that traps provide a superior means of replicating a collecting technique and making quantitative comparisons among sites (*Ramalho et al., 2013*; *Sydney & Gonçalves, 2015*).

Third, for logistical reasons, we did not sample certain habitats in Belize, including low elevation sites in the northeast (Corozal District), undeveloped rainforests of Temash/ Sarstoon National Park in the southeast, and high elevation areas of Belize that reach over 1,100 m. We found no correlation of elevation with either species richness or diversity, but that reflects a range of elevations of just 15–600 m. *Roubik & Hanson (2004)* note that the diversity of orchid bees peaks at ~800 m elevation in southern Central America, and that up to a point, *Eulaema* and *Eufriesea* increase in diversity as elevation increases. However, they also suggest that bees found at high elevations may have home ranges encompassing both low and high elevations, especially because orchid bees can fly long distances over short periods of time (*Witelski et al., 2010*; *Pokorny et al., 2015*).

Fourth, we sampled only October–March, though our sample dates included both the late-wet (Dec–Jan) and early-dry (Feb) seasons (*Climate Change Knowledge Portal, 2018*). This interval spanned a good part of the range of seasonal climatic variation in Belize, but we may have missed species inactive as adults or lower in abundance due to the well-documented seasonality of some orchid bees (*Janzen et al., 1982*; *Ackerman, 1983b*; *Roubik & Ackerman, 1987*; *Knoll, 2016*; *Nemésio, Santos & Vasconcelos, 2015*; *Costa & Francoy, 2017*; *Margatto et al., 2019*; *Bravo et al., 2022*). However, most euglossines are active as adults through most of the year (*Roubik & Hanson, 2004*; see also *Janzen et al., 1982*), and *Ackerman (1983b)* found that species composition and relative abundance at lowland sites in Panama were comparable across seasons. Thus, most species should be found with intensive sampling even outside of their seasonal peaks, with the exception of

*Eufriesea*, whose adults are often active during restricted times the year (*Roubik & Hanson, 2004*). At least seven species of *Eufriesea* that we did not collect are known to occur at latitudes both north and south of Belize (Table S9). Collection records for *Eufriesea* often give dates outside of our primarily December–February sampling periods (*Janzen et al., 1982*; *Kimsey, 1982*; *Ascher & Pickering, 2019*).

## CONCLUSIONS

To our knowledge, our study represents the first geographically wide-ranging and quantitative assessment of orchid bees within Belize. Related to our objectives, our trap-based surveys documented the presence in Belize of 24 species that varied in relative abundance and distribution in relation to environmental gradients. However, a Chao1 analysis and inspection of records of species in nearby countries indicate that further species will be found. Because our study was designed primarily as a broad appraisal of the orchid bees present during the late-wet and early-dry seasons, the geographic and abundance patterns (especially their link to agricultural activities) that we found must be considered tentative. However, our repeated sampling at some locations and sites indicates that orchid bee species are not randomly or evenly distributed in the country, with a clear gradient in species composition of samples from north to south (paralleling a precipitation gradient). Future surveys in Belize would benefit from being conducted during other times of year, especially during the June–September wet season, as well as in several major locations we did not sample, in order to extend the range of variation in elevations, habitat classes, and quantified levels of disturbance. Combined with surveys of other insects, future studies of euglossines in Belize could contribute to determining the value of orchid bees as overall indicator species in conservation studies, which has been the subject of recent debate (*Añino, Parra & Galvez, 2019*; *Allen et al., 2019*; *Miranda et al., 2019*).

## ACKNOWLEDGEMENTS

We thank Shanelly Carillo, Victoria Chi, Edgar Correa, Maria del Carmen Flores, and Liborio Santos of the Belize Forest Department for assistance in obtaining Research and Collection and Export Permits. Samuel O'Neill and Thomas O'Neill assisted with field work. Dr. Jerry Rozen and Corey Smith of the American Museum of Natural History and Dr. Terry Griswold of the USDA-ARS, Logan, Utah loaned us reference specimens of orchid bees; Dr. James Leibherr provided access to the Cornell University Insect Collection. We express our gratitude for the invaluable assistance of the many Belizeans who gave advice on local sampling sites, provided hospitality, and gave permission to conduct research on their lands: The Belize Audubon Society, The Friends for Conservation and Development, The Programme for Belize. Cameron and Kelly Boyd, Victor Cho, Denham Chuc, Arvin Coc, Carlos Corrales, Diego Cruz, Nicacio Coc, Giovanni and Jenny Fernandez, Eduardo and Allison Gonzalez, Mike and Jenny Hall, Derick Hendry, Rafael Mesh, Isaias Morataya, Ian and Kate Morton, Fernando Obando, Peter and Petro Steunenberg, Israel Pau, Simon Pau, Jose Perez, Fredy Pineda, Victor Pixabaj, Joy Smith, Ray and Vicki Snaddon, Thyra Thompson, Melvis Valdez, Fedrito Villaneuva, and Mick and Angie Webb. Special thanks go to Dominique Lizama of the

Belize Audubon Society, to Ian Morton who gave us our first hands-on demonstration of attracting orchid bees with baits, and to the rangers of the Chiquibul Forest who delivered us to the Las Cuevas Research Station in a truck lacking brakes. We thank Dr. Tracy Sterling, Melody Schimpf, and Ana Murphy of the Department of Land Resources and Environmental Sciences, Montana State University for their great help in organizing our trips to Belize.

### Funding

The research was supported with funds from the Montana Agricultural Experiment Station, Montana State University and the Montana State University Faculty Excellence Program. The funders had no role in study design, data collection and analysis, decision to publish, or preparation of the manuscript.

### Grant Disclosures

The following grant information was disclosed by the authors:
Montana Agricultural Experiment Station, Montana State University and the Montana State University Faculty Excellence Program.

### Competing Interests

The authors declare that they have no competing interests.

### Author Contributions

- Kevin M. O'Neill conceived and designed the experiments, performed the experiments, analyzed the data, prepared figures and/or tables, authored or reviewed drafts of the article, and approved the final draft.
- Ruth P. O'Neill conceived and designed the experiments, performed the experiments, analyzed the data, prepared figures and/or tables, authored or reviewed drafts of the article, and approved the final draft.
- Casey M. Delphia conceived and designed the experiments, performed the experiments, analyzed the data, prepared figures and/or tables, authored or reviewed drafts of the article, and approved the final draft.
- Laura A. Burkle conceived and designed the experiments, analyzed the data, prepared figures and/or tables, authored or reviewed drafts of the article, and approved the final draft.
- Justin B. Runyon conceived and designed the experiments, performed the experiments, authored or reviewed drafts of the article, and approved the final draft.

### Field Study Permissions

The following information was supplied relating to field study approvals (*i.e.*, approving body and any reference numbers):

Collecting permits were granted by the Forestry Department, Belize Ministry of Sustainable Development, Climate Change and Disaster Risk Management (Belize

Forestry Department Permit Reference Numbers: CD/60/3/15(43); WL/1/1/16(52); WL/2/1/18(38); FD/WL/1/19(33); CD/60/3/15(25); WL/2/1/17(05); WL/2/1/18(32)).

## Data Availability

The raw data on distribution and abundance of the orchid bees, as well as the environmental data, is available in the Supplemental Files.

## Supplemental Information

Supplemental information for this article can be found online at http://dx.doi.org/10.7717/peerj.14928#supplemental-information.

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
