# Peer review of "Diversity and distribution of orchid bees (Hymenoptera: Apidae, Euglossini) in Belize"

_PeerJ, doi:10.7717/peerj.14928_

## Round 0.1 · original submission · Major Revisions

Dear Authors

Thank you for your submission. The manuscript is evaluated by three reviewers and all found major revision required before consideration. Major issues were found in the description of methods, results and text write-up. Authors must address each comments of all reviewers in the revised version.

Reviewer 1 ·

Basic reporting

no comment.

Experimental design

sem comentários.

Validity of the findings

The paper presents important and interesting results about the Euglossini bee fauna of Belize.

Additional comments

I am suggesting a major revision of the manuscript because it can be beneficed by just stressing the main question of the research and rewriting all the manuscript having it in mind. Other suggestions are in the manuscript.

I hope that with a revision of the manuscript, the paper will interest to a much wider audience.

Annotated reviews are not available for download in order to protect the identity of reviewers who chose to remain anonymous.

Reviewer 2 ·

Basic reporting

OK. The manuscript is well written.

Experimental design

My main concern is related to species identification, which was not confirmed by a specialist in Euglossini taxonomy. Although, the authors identified the species using different known references and by comparison with reference material in Zoological Collections. In my opinion is imperative to send samples of morphotypes or specimens identified until species level to a specialist. Despite this, there is always a risk of mistakes in species identification, since many similarities between different species are usual in this group of bees. Another concern is related to the use of different sets of chemical fragrances (ranging from 3 to 9) through the different periods of samplings.

Validity of the findings

This is a relevant contribution, however I deeply recommend to authors to send samples of the identified material to be checked and confirmed by a taxonomist (with expertise in Euglossini bees).

Additional comments

MS-Peer J #76014
Diversity and distribution of orchid bees (Hymenoptera: Apidae, Euglossini) in Belize

This is an original paper on euglossine bees, which describes the species richness and diversity of assemblages of this group of Neotropical bees in different localities in Belize. The study improves the current knowledge of orchid bees across Central America. Using bait-traps, with different set of chemicals, the authors captured 4,571 male orchid bees, distributed in 25 orchid bee species, in four genera. Samplings were carried out during the late-wet and early-dry seasons of 2015-2020, distributed in four periods. However, despite being an important contribution on the Euglossini knowledge, I have some concerns regarding bee methodology and result presentation. My main concern is related to species identification, which was not confirmed by a specialist in Euglossini taxonomy. Although, the authors identified the species using different known references and by comparison with reference material in Zoological Collections. In my opinion is imperative to send samples of morphotypes or specimens identified until species level to a specialist. Despite this, there is always a risk of mistakes in species identification, since many similarities between different species are usual in this group of bees. Another concern is related to the use of different sets of chemical fragrances (ranging from 3 to 9) through the different periods of samplings. The presentation of the results is in some way confusing to the reader. For instance, there are four Tables (I to IV) showing the dates of samplings in each locality per site and per period (year), while the lists of species, with information on abundance and other relevant information were given as supplementary material. I suggest to authors to include the measures of species Below, I have listed some concerns, suggestions and questions needing be addressed before publication.
Concerns and recommendations:
INTRODUCTION section
1 – L.67-68, 70 etc – replace Moure, Melo & Faria (2012) by Moure & Melo (2022): http://moure.cria.org.br/catalogue
(How to cite this page
J. S. Moure & G. A. R. Melo, 2022. Euglossini Latreille, 1802. In Moure, J. S., Urban, D. & Melo, G. A. R. (Orgs). Catalogue of Bees (Hymenoptera, Apoidea) in the Neotropical Region - online version. Available at http://www.moure.cria.org.br/catalogue. Accessed Sep/21/2022)
2 – L.93- replace Oliviera by Oliveira
METHODS
3 – L. 189-192 - This is a point of concern. The absence of any taxonomic confirmation carried out by a specialist with expertise in the taxonomy of orchid bees.–
4 – L.251 and across the MS – e.g., Eug. purpurea, Eug. ignita, Eul. marcii, Eug. mixta, and Eug. imperialis
It is unusual in orchid be studies use the abbreviation of the name of genera as the authors used in their paper. Replace Eug. by Eg., Eul. by El. and Euf. by Ef. Across the text and the Tables
RESULTS
5 – L. 484. A more precise information in approximately 250 species (see Moure & Melo, 2022)
REFERENCES
Moure & Melo, 2022
L. 711 – Ribeiro instead Ribiero

Reviewer 3 ·

Basic reporting

Abstract- The abstract totally ignores pollination of many non-orchids. What do the correspondence analyses add to the Hull/Simpson type?
Neither gradients or elevations are clear.
Reasonably complete in Methods but vague considering why this particular study guides us to a firmer stance in anything in particular.

No references are made to the big field data analysis by Ramirez et al. 2001. Acta Biologica Colombiana, or to two expressly long-term studies of Central American euglossines, one by Roubik et al. 2021, published in Conservation Science and Practice, and one by P. E. Hanson, as a coauthor, on bee species present over 40 years in part of Costa Rica..

Should use abbreviations for genera used widely, of other pubs- Ex. El. Ef. Eg.

279, 445 Euglossa cyanura desc. From Panama is more than a single species, not found N of S. CR.
445 Eulaema luteola, not Euglossa luteola, common in S. Mex
Eulaema marcii is probably El. cingulata, as given in R&H 2004. There is strong indication that the El. marcii of SE Brazil is rather endemic, while there is still no robust demonstration that El. cingulata and El. pseudocingulata both live in French Guiana.
Furthermore, from all the baiting in Costa Rica and Panama, El. cingulata never arrives at cineole. So, it is either an undescribed sp. or (more likely) it is taking advantage of scent resources unused in Belize by El. nigrita, which does come often to cineole and has a N. range limit of Costa Rica.

Eufriesea tend to be seasonal and submontane or montane.

Experimental design

It seems to me that more indications of actual elevation and rainfall are needed. They are not spelled out. Variation is beyond the scope of the data analysis, but should be included (Roubik et al., 2021, ref mentioned above).

For some reason, a whole paragraph is devoted to supposed 'canopy versus lower' comparisons. The cited paper from 2022 does not even approach the canopy, only 12 m. A much earlier paper looked at it over a whole year, and included 27 m in height, and should be the reference point. It was from Panama (see Roubik, J. Insect Behavior, 1991). Furthermore, there is no reason to establish that bees fly equally up high or down low- they mostly avoid intense heat and sunlight, but can dodge around. Sampling in systematic surveys merely require repeated measures, and that is sufficient for the purposes of this study. What are those purposes? It seems, only at the end of the discussion, that they boil down to having a good reason to bait and kill thousands of bees that come to chemical baits.

The authors have chosen to shift around their attractant type and number, exposing the whole study to 'flaws'. I, for one, do not care what attractants they use, if they are not trying to demonstrate trends or qualities over time. They try to compare different habitats but, without real seasonality data, they seem to this reviewer to never have had a chance. Better to stick to actual species reports. So what, then, is the difference in taking more data from Belize, versus looking at big lists from Costa Rica, Panama and Mexico and looking for any possibility of endemism in Belize? I don't see it.

It is also rather alarming that 10, 17 or 44 CONSECUTIVE DAYS of baited traps produced the data. How can these possibly be compared, when it is very likely that some species are depleted more rapidly than others, and that larger bees, having larger flight ranges and longer tongues, are best able to circulate and forage widely?

Validity of the findings

Discussion. Do we really know much from a survey that included a mean of 9 spp. in sites that all should have 20 or more common species and upwards of 35-40? How far do the data actually go, about bees that can easily fly dozens of km, in any kind of matrix, including some which include some agriculture but, by the same token, include many more open habitats, which may encourage presence of many more nectar and some pollen species (like roadside Melastomataceae, Solanaceae, Passifloraceae, etc.). Floral resource data are still very minimal in the literature, i.e. we still are too unfamiliar with what it takes to grow orchid bee populations. How can a little data base from a little country be that useful? BTW, Ex. dentata, a very rare euglossine, has been collected there- SJ pers. comm. To Roubik, in a paper by that author, in Psyche, 2019.

Additional comments

The Eulaema cingulata issue is this: There are marcii, pseudocingulata and cingulata. Genitalic characters seem to suggest the three are different. DNA suggests two are the same. El. marcii seems to be endemic to SE Brazil and all others, except those from French Guiana, can be called El. cingulata, until someone does some better work on this problem. The Eg. cyanura problem is more gnarly. Hinojosa Diaz and Engel missed it in their extensive Euglossa work. There is still a lot of basic work to do. And one of the more basic survey questions is to either make a big comparison of a lot of sites at more or less the same time (e.g. B. Brosi and his paper), or go for a long-term approach, with replicates in relatively similar habitats. Collecting a pile of bees is quite easy. Getting to where we want to go is another thing. And moderating supposedly controversial topics which are largely based on incomplete or biased data is, in my opinion, largely a waste of time.

---

## Round 0.2 · Minor Revisions

Dear Authors
Thank you for your submission again. The reviewers have some minor concerns still pending. The authors should clear it before the final decision is reached. The comments of both reviewers are attached herewith.

Reviewer 2 ·

Basic reporting

Minor carried was out in the text in this new version. In general, the text is well written. The structure of the paper is ok.

Experimental design

no comment here

Validity of the findings

The findings are important to improve our knowledge on orchid bees through Neotropics.

Additional comments

Authors answered to my questions with strong argumentation. Although I do not entirely agree with their responses to my questions and I do not feel completely convinced with their arguments, since the study add new and relevant information on euglossine bees through Neotropics, I recommend the paper to publication.

Reviewer 3 ·

Basic reporting

Enough is shown to warrant a preliminary study that really went beyond a short term snapshot, and is appreciated.

Experimental design

The limitations are clearly given, but it is still not entirely clear how many days different places were sampled under different circumstances or seasons. It seems, in the end, that only very broad comparisons, with only basic and tentative conclusions, can be made. This seems reasonable, and certainly has its place, especially in Belize, which has differing climates and also a real history of large hurricane disturbances.

Validity of the findings

These are broad and clear, with a lot of caveats for anything else. Again, this now seems more reasonable.

Additional comments

A little over reacting by a reviewer to two puts any authors, or the study, to a test. Excuse the assumption, but I thought it was perhaps useful. Some of the taxonomy is still up in the air, but that should be left to the taxonomists- when they get around to it.

---

## Round 0.3 · accepted · Accept

Dear Authors

It is pleasure to inform you that the manuscript has been accepted. The authors have addressed the reviewers' comments and now it is ready for publication.